# Dual Inhibition of H3K9me2 and H3K27me3 Promotes Tumor Cell Senescence without Triggering the Secretion of SASP

**DOI:** 10.3390/ijms23073911

**Published:** 2022-04-01

**Authors:** Na Zhang, Mengjie Shang, Hongxin Li, Lan Wu, Meichen Dong, Baiqu Huang, Jun Lu, Yu Zhang

**Affiliations:** 1The Key Laboratory of Molecular Epigenetics of Ministry of Education (MOE), Northeast Normal University, Changchun 130024, China; zhangn783@nenu.edu.cn (N.Z.); lihx744@nenu.edu.cn (H.L.); dongmc997@nenu.edu.cn (M.D.); huangbq@nenu.edu.cn (B.H.); 2The Institute of Genetics and Cytology, Northeast Normal University, Changchun 130024, China; shangmj340@nenu.edu.cn (M.S.); wul593@nenu.edu.cn (L.W.); luj809@nenu.edu.cn (J.L.)

**Keywords:** tumor cell senescence, SASP, CCF, H3K9me2, H3K27me3

## Abstract

Chemotherapy remains the most common cancer treatment. Although chemotherapeutic drugs induce tumor cell senescence, they are often associated with post-therapy tumor recurrence by inducing the senescence-associated secretory phenotype (SASP). Therefore, it is important to identify effective strategies to induce tumor cell senescence without triggering SASP. In this study, we used the small molecule inhibitors, UNC0642 (G9a inhibitor) and UNC1999 (EZH2 inhibitor) alone or in combination, to inhibit H3K9 and H3K27 methylation in different cancer cells. Dual inhibition of H3K9me2 and H3K27me3 in highly metastatic tumor cells had a stronger pro-senescence effect than either inhibitor alone and did not trigger SASP in tumor cells. Dual inhibition of H3K9me2 and H3K27me3 suppressed the formation of cytosolic chromatin fragments, which inhibited the cGAS-STING-SASP pathway. Collectively, these data suggested that dual inhibition of H3K9 and H3K27 methylation induced senescence of highly metastatic tumor cells without triggering SASP by inhibiting the cGAS-STING-SASP pathway, providing a new mechanism for the epigenetics-based therapy targeting H3K9 and H3K27 methylation.

## 1. Introduction

In recent years, several strategies for the treatment of cancer have been developed; however, chemotherapy remains the preferred strategy for the treatment of multiple tumors. Tumor cell senescence induced by chemotherapeutic drugs is as an important cytological mechanism for the inhibition of tumor [1]. However, tumor recurrence occurs frequently after exposure to chemotherapeutic drugs. This can be related to the senescence-associated secretory phenotype (SASP) [2,3,4]. SASP is a two-edged sword that may both prevent tumor formation and trigger immune clearance of senescent cells from tissues by producing the senescent phenotype in both an autocrine and paracrine way [5,6]. However, SASP also plays atumorigenic role by promoting cell growth and the epithelial-mesenchymal transition (EMT) of tumor cells [7,8]. Therefore, an effective strategy that induces tumor cell senescence without triggering SASP may reduce the rate of tumor recurrence after chemotherapy and provide new ideas for tumor treatment.

Cytoplasmic chromatin fragments (CCF) are associated with cellular senescence [9,10,11]. In senescent cells, CCF is formed when nuclear membrane blebs containing chromatin fragments develop and subsequently divide into the cytoplasm [12,13]. CCF contain genomic DNA, γH2AX, and the heterochromatin markers, H3K9me3 and H3K27me3 [12]. The cytosolic DNA sensor, cGAS, recognizes cytoplasmic DNA and produces cyclic GMP-AMP (cGAMP), which activates STING [14,15]. The pathway plays important roles in triggering inflammation, and the inflammatory cytokinesare also referred to as SASP [8,16,17,18]. CCF accumulate in senescent cells induced by oxidative stress through activating cGAS-STING-NF-κB signaling, thereby inducing SASP [19]. These findings add to the evidence for CCF-cGAS-STING signaling’s pro-senescent role and the molecular foundation of SASP regulation. However, it remains unknown whether CCF play similar roles in other types of senescence.

Accumulating evidence suggests that epigenetic mechanisms, such as histone modification, are associated with the pathogenesis of multiple cancers; therefore, targeting histone modification maybe a promising therapeutic strategy for cancer [20]. Methylation of H3K9 and H3K27, which is associated with gene silencing is involved in multiple myeloma [21,22,23,24]. Small molecule inhibitors of EZH2 (a H3K27 methyltransferase) have antimyeloma effects [22,24], and inhibitors of G9a (a H3K9 dimethyltransferase, also called EHMT2) have anticancer effects against urinary bladder cancer cells [25]. Thus, methylation of H3K9 and H3K27 could begreat therapeutic targets for cancer treatment. EZH2 and G9a work togetherto suppress gene expressionin human fibroblasts and mouse embryonic stem cells [26,27]. Furthermore, dual G9a and EZH2 inhibition reduces myeloma cell proliferation by modulating the interferon signal and the IRF4-MYC axis [28]. However, the mechanism underlying dual G9a and EZH2 inhibition remains unclear. Moreover, the effect of controlling the formation of CCF by targeting H3K9me3 and H3K27me3, which are the important components of CCF, is unknown.

In the present study, we revealed a novel mechanism underlying tumor cell senescence following epigenetics-based therapy whereby dual inhibition of H3K9me2 and H3K27me3 promoted tumor cell senescence without triggering secretion of SASP. These results supported the use of histone methyltransferase inhibitors as a promising epigenetics-based therapy.

## 2. Results

### 2.1. Dual Inhibition of H3K9me2 and H3K27me3 Inhibits the Proliferation and Migration of Highly Metastatic Tumor Cells

H3K9 and H3K27 methylation inhibits gene transcription and plays essential roles in the pathogenesis of multiple cancers [29,30,31,32]. Dual inhibition of G9a (H3K9 methyltransferase) and EZH2 (a H3K27methyltransferase) exerts a strong antitumor effect in myeloma [28]. The role of dual G9a and EZH2 inhibition in other cancers remains unclear. First, we analyzed the expression of G9a and EZH2 in breast, colon, and prostate cancers, according to The Cancer Genome Atlas (TCGA) database. The results showed that G9a and EZH2 were expressed at markedly higher levels in primary tumor samples than in normal tissue samples (Appendix A). Therefore, we initially compared the expression level of EZH2 and G9a in noncancerous mammary epithelial cells (MCF10A) and multiple cell lines from breast cancer (MCF7, BT549, and MDA-MB-231), colon cancer (SW480 and HCT116), and prostate cancer (PC3 and DU145). The expression of EZH2 and G9a was considerably higher in breast cancer cells (MCF7, BT549, and MDA-MB-231) than in the noncancerous mammary epithelial cell (MCF10A), especially in the highly metastatic cells of the MDA-MB-231 cell line (Figure 1A,B, left). The expression of EZH2 and G9a was also higher in highly metastatic colon cancer cells (HCT116) or prostate cancer cells (DU145) than in SW480 and PC3 cells (Figure 1A,B right). This suggested that EZH2 and G9a were associated with tumor malignancy. Next, we detected the methylation levels of H3K9 (H3K9me2 and H3K9me3) and H3K27 (H3K27me3) in these cell lines. The methylation levels of H3K9 (H3K9me2 and H3K9me3) and H3K27 (H3K27me3) were remarkably higher in highly metastatic cell lines (MDA-MB-231, HCT116, and DU145) (Figure 1C), suggesting that H3K9 and H3K27 methylation was involved in malignant cancer progression. To determine the effect of dual inhibition of H3K9 and H3K27 methylation, tumor cell lines were treated with the G9a inhibitor, UNC0642 (1 µM), and the EZH2 inhibitor, UNC1999 (5 µM), or UNC0642 (1 µM) +UNC1999 (5 µM) for seven days (Appendix A). Dual inhibition of H3K9me2 and H3K27me3 inhibited the proliferation and migration of tumor cells with high expressions of H3K9me2 and H3K27me3 (Figure 1D and Appendix A). These findings suggested that dual inhibition of H3K9me2 and H3K27me3 inhibited the proliferation and migration of highly metastatic tumor cells.

### 2.2. Dual Inhibition of H3K9me2 and H3K27me3 Has a Stronger Pro-Senescence Effect than Single Inhibition in Highly Metastatic Tumor Cells

Initial experiments (Figure 1) showed a dramatic inhibition of tumor cell proliferation in response to dual inhibition of H3K9me2 and H3K27me3. We, therefore, analyzed the effect of UNC0642 and UNC1999 alone or in combination on the cell cycles progression. Flow cytometry analysis of the DNA content showed that dual inhibition of H3K9me2 and H3K27me3 increased the number of MDA-MB-231 cells in the G1 phase (Figure 2A). Cells arrested in the G1phase (G1 tetraploid cells) are in replicative or oncogenic Ras-induced senescence [33,34]. Senescent G1 tetraploid cells exhibit a dramatic downregulation of CyclinA2 [35]. To determine whether dual inhibition of H3K9me2 and H3K27me3 causes MDA-MB-231 cells to enter cellular senescence and undergo G1 arrest, we examined the expression of CyclinA2, the activity of senescence-associated β-galactosidase (SA-β-gal), and the expression of Ki67. The results showed that dual inhibition of H3K9me2 and H3K27me3 had a stronger pro-senescence effect than inhibition of either enzyme alone in breast cancer cells (MDA-MB-231), as indicated by the downregulation of CyclinA2 (Figure 2B,C), increased activity of SA-β-gal (Figure 2D,E), and a decrease in Ki67-positive cells (Figure 2F,G). Similar results were observed in two other cell lines with the indicated treatments (HCT116 and DU145) (Appendix A). However, in cells with a low degree of malignancy (MCF7, SW480, and PC3), dual inhibition of H3K9me2 and H3K27me3 did not significantly inhibit the expression of CyclinA2 under the same conditions (Appendix A). Increased exposure to inhibitors of H3K9me2 and H3K27me3 promoted MCF7 cellular senescence, as indicated by the downregulation of CyclinA2 (Appendix A) and increased activity of SA-β-gal (Appendix A). These results indicated that highly metastatic tumor cells were more sensitive to the inhibitors, UNC0642 and UNC1999. Taken together, the results suggested that dual inhibition of H3K9me2 and H3K27me3 promoted cellular senescence in highly metastatic tumor cells, and the effect was stronger than that of single inhibition.

### 2.3. Dual Inhibition of H3K9me2 and H3K27me3 Induces Tumor Cell Senescence without Triggering SASP

Our initial experiments showed that dual inhibition of H3K9me2 and H3K27me3 promotes tumor cell senescence (Figure 2). However, studies show that tumor cell senescence induced by chemotherapeutic drugs also causes tumor recurrence by inducing SASP, which promotes cell growth and EMT [4,7,8]. Thus, we assessed the effect of dual inhibition of H3K9me2 and H3K27me3 on SASP in senescent tumor cells and the difference between senescence induced by dual inhibition and that induced by chemotherapeutic drugs. For this purpose, we used the chemotherapeutic drug, doxorubicin, which may induce tumor cell senescence and trigger the secretion of SASP [36,37], as the positive control. Both doxorubicin and dual inhibition of H3K9me2 and H3K27me3 induced senescence of breast cancer cells (MDA-MB-231), accompanied by downregulation of CyclinA2 (Figure 3A and Appendix A). Dual inhibition of H3K9me2 and H3K27me3 inhibited SASP, as indicated by a decrease in SASP components, IL-6 and IL-8, in senescent breast cancer cells. This differed from the effect of doxorubicin, which promoted SASP by increasing of IL-6 and IL-8 (Figure 3A). The same results were observed in colon cancer cells (HCT116) and prostate cancer cells (DU145) with the indicated treatments (Figure 3B,C). Taken together, these data indicated that dual inhibition of H3K9me2 and H3K27me3 induced tumor cell senescence without triggering SASP.

### 2.4. Dual Inhibition of H3K9me2 and H3K27me3 Suppresses cGAS-STING Signaling by Decreasing CCF

We confirmed that dual inhibition of H3K9me2 and H3K27me3 induced tumor cell senescence without triggering SASP, which differed from the effect of doxorubicin (Figure 3). However, the reason for this difference remained unclear. Damaged chromatin fragments are released from the nucleus into the cytoplasm to form CCF during the tumor cell senescence [12,13], which promotes SASP. Because the heterochromatin markers, H3K9me3 and H3K27me3, are components of CCF, we speculated that dual inhibition of H3K9me2 and H3K27me3 may inhibit the formation of CCF. To test this hypothesis, CCF formation in tumor cells treated as indicated was assessed by immunofluorescence. The results showed that the number of CCF-positive cells increased significantly in senescent breast cancer cells (MDA-MB-231) treated with doxorubicin (Figure 4A), whereas dual inhibition of H3K9me2 and H3K27me3 decreased the proportion of CCF-positive cells or even inhibited them (Figure 4B) in breast cancer cells (MDA-MB-231). This indicated that dual inhibition of H3K9me2 and H3K27me3 inhibited the formation of CCF, thereby blocking SASP.

Although CCF promote SASP in senescent cells [12], they do not act on the production of SASP. When the chromatin fragments are released into the cytoplasm, CCF are recognized directly by cGAS and activate cGAS-STING signaling, resulting in the production of SASP [19]. Because our initial results showed that dual inhibition of H3K9me2 and H3K27me3 inhibited the formation of CCF (Figure 4B), we tested whether the CCF-induced SASP depended on cGAS-STING signaling. The results showed that dual inhibition of H3K9me2 and H3K27me3 suppressed cGAS-STING signaling significantly, as indicated by decreased p-STING, IL-6, and IL-8 (Figure 4C). Exogenous expression dnMCAK (a promoter of CCF) [38] in the breast cancer cells treated with H3K9me2 and H3K27me3 inhibitors increased the formation of CCF and suppressed the reduction of SASP and inhibition of cGAS-STING signaling mediated by dual inhibition of H3K9me2 and H3K27me3, as indicated by increased p-STING, IL-6, and IL-8 levels (Figure 4D). Taken together, the results indicated that dual inhibition of H3K9me2 and H3K27me3 suppressed cGAS-STING signaling by decreasing CCF in breast cancer cells, thereby inhibiting SASP.

## 3. Discussion

In this study, we found that dual inhibition of H3K9me2 and H3K27me3 using a combination of UNC0642 and UNC1999 exerted a stronger pro-senescence effect in malignant cancer cells than single inhibition. Dual inhibition of H3K9me2 and H3K27me3 suppressed SASP by decreasing the formation of CCF through a mechanism involving cGAS-STING signaling (Figure 5).

Epigenetic mechanisms such as histone modification are associated with human diseases, especially cancer [20]. The methylation levels of H3K9 modulated by G9a and H3K27 modulated by EZH2 are often increased in cancer and associated with tumor progression [29,39,40]. Inhibition of G9a by the small molecule inhibitor, UNC0642, significantly decreases cell viability in melanoma cells [41] and induces apoptosis of human bladder cancer cells [25]. Inhibition of EZH2 by the small molecule inhibitor, UNC1999, suppresses bladder cancer cell growth and metastasis [42]. In addition, dual inhibition of H3K9me2 and H3K27me3 with the G9a inhibitor, UNC0638 (1 μM), and the EZH2 inhibitor, GSK126 (1 μM), suppresses multiple myeloma cell proliferation by inducing cell cycle arrest and apoptosis [28]. This suggests that H3K9 and H3K27 methylation is a potential target for cancer therapy. However, the effect of dual G9a and EZH2 inhibition in other cancers and the underlying mechanisms remain unclear. The present study is the first to show that dual inhibition of H3K9me2 and H3K27me3 using a combination of UNC0642 and UNC1999 induced the senescence of highly metastatic tumor cells, and the effect was stronger than that of either inhibitor alone (Figure 2 and Appendix A). Compared with highly metastatic tumor cells (MDA-MB-231, HCT116, and DU145), cells with a low degree of malignancy (MCF7, SW480 and PC3) were insensitive to dual inhibition of H3K9me2 and H3K27me3 under the same treatment conditions, as indicated by the lack of significant changes in the expression of CyclinA2 (Appendix A). This suggested that drug sensitivity was related to tumor malignancy. The use of the G9a inhibitor and the EZH2 inhibitor for dual inhibition of H3K9me2 and H3K27me3 may be more suitable for the treatment of tumors with a high degree of malignancy, stronger migration ability, and high expression of H3K9me2 and H3K27me3.

Cell senescence is a relatively stable state of cell cycle arrest and an important cytological mechanism for the prevention of cancer [43]. Tumor cell senescence is associated with compromised integrity of the nuclear envelope [9,10,11]. Recent studies reported the presence of CCF during senescence [12,13]. CCF activate the cytosolic DNA-sensing cGAS-STING pathway, thereby promoting SASP in primary human cells [14,15,16]. In this study, dual inhibition of H3K9me2 and H3K27me3 suppressed cGAS-STING signaling by decreasing CCF in tumor cells, thereby inhibiting the secretion of SASP (Figure 2). However, tumor cells remained senescent under dual inhibition of H3K9me2 and H3K27me3 using a combination of UNC0642 and UNC1999 (Figure 2 and Appendix A). This difference may be attributed to different types of cellular senescence induced by oncogenic HRasV12, DNA damage, or replication exhaustion. However, the mechanisms underlying tumor cell senescence induced by dual inhibition of H3K9me2 and H3K27me3 remain unclear. Studies show that inhibition of EZH2/G9a upregulates interferon (IFN)-stimulated genes and suppresses IRF4-MYC axis genes in multiple myeloma [28]. IFNγ and TNF cooperate to upregulate the expression of p16Ink4a and p21Cip1 to promote tumor cell senescence through the JAK-STAT1 signaling pathways [44]. Downregulation of EZH2 decreases the levels of H3K27me3, displaces BMI1, and activates p16Ink4a transcription, which promotes the tumor cell senescence [45]. Whether the IFNγ-JAK-STAT1-p16Ink4a axis was involved in tumor cell senescence induced by dual inhibition of H3K9me2 and H3K27me3 remains to be explored.

Chemotherapy can increase intratumoral heterogeneity, leading to different fates of tumor cells: senescence or apoptosis [46]. Many genotoxic chemotherapies target proliferating cells nonspecifically, often with adverse reactions. Doxorubicin, cisplatin, bleomycin, and other chemotherapy drugs can induce cancer cellular senescence accompanied by SASP [47]. SASP is related to increased tumor incidence and tumor recurrence [7]. Therefore, the identification of an effective chemotherapeutic drug capable of inducing tumor cell senescence without triggering SASP is important. In this study, compared with doxorubicin, dual inhibition of H3K9me2 and H3K27me3 with the G9a inhibitor, UNC0642, and the EZH2 inhibitor, UNC1999, promoted tumor cell senescencewithout triggering SASP (Figure 3). This suggested that the combination of anEZH2 inhibitor and a G9a inhibitor may be more suitable for the treatment of cancer than doxorubicin and may decrease the risk of tumor recurrence. In addition, doxorubicin is an anthracycline antibiotic with strong anticancer activity that is commonly used for cancer treatment. A combination of use of doxorubicin, UNC0642, and UNC1999 may be an effective therapeutic strategy for the treatment of tumors.

## 4. Materials and Methods

### 4.1. Cell Cultures

Cell lines (HEK293T, MCF10A, MCF7, BT549, MDA-MB-231, SW480, HCT116, PC3, and DU145) were obtained from the American Type Culture Collection (ATCC, Manassas, VA, USA). These cells were characterized by DNA fingerprinting and isozyme detection and were tested with a MycoBlue Mycoplasma Detector (Vazyme Biotech, Nanjing, China) to exclude mycoplasma contamination before the experiments. MCF10A cells were cultured in DMEM-F12 supplemented with 5% (*v*/*v*) horse serum, 20 ng/mL EGF, 0.5 mg/mL hydro cortisone, 100 ng/mL cholera toxin, 10 mg/mL insulin, and 10 U/mL penicillin-streptomycin. MDA-MB-231 cells were cultured in L-15 medium (Sigma-Aldrich, St. Louis, MO, USA) with 10% (*v*/*v*) fetal bovine serum (VivaCell, Shanghai, China). HEK293T, MCF7, BT549, SW480, HCT116, PC3, and DU145 cells were cultured in DMEM (Sigma-Aldrich) with 10% (*v*/*v*) fetal bovine serum (VivaCell, Shanghai, China). All cell lines were grown at 37 °C with 5% CO_2_ except MDA-MB-231, which was cultured at 37 °C without CO_2_.

### 4.2. Reagents and Drug Treatment

UNC1999 was purchased from MCE (MedChemExpress HY-15646). UNC0642 was purchased from MCE (MedChemExpress, HY-13980). Doxorubicin (Dox) was obtained from Selleck (S1208). Cell lines were treated with a single inhibitor (UNC1999 or UNC0642) or a combination of the two inhibitors (UNC1999 + UNC0642) or with DMSO for up to seven days (MDA-MB-231, HCT116, DU145) or nine days (MCF7, SW480, PC3), refreshing the medium and drugs every three days.

### 4.3. Plasmid and Retroviral Infection

The pCDH-CMV-Flag-dnMCAK expression vector [48], and the packaging vectors, psPAX2 and pMD2.G, were used in this study. The generation of lentivirus in HEK293T cells and the transfection of lentiviral constructs into recipient cells were performed according to the manufacturer’s instructions (Invitrogen, Carlsbad, CA, USA).

### 4.4. Reverse Transcription, PCR and Real-Time PCR

Reverse transcription, PCR, and real-time PCR were performed, as described previously [49]. The primer sequences for PCR were as follows (5′-3′, sense, antisense): β-actin forward: 5′-GAGCACAGAGCCTCGCCTTT-3′ and reverse: 5′-ATCCTTCTGACCCATGCCCA-3′; EZH2 forward: 5′-ATGGGCCAGACTGGGAAGAA-3′ and reverse: 5′-TCAAGGGATTTCCATTTCTCT-3′; G9a forward: 5′-CTGTCAGAGGAGTTAGGTTCTGC-3′ and reverse: 5′-CTTGCTGTCGGAGTCCACG-3′; CyclinA2 forward: 5′-TTCATTTAGCACTCTACACAGTCACGG-3′ and reverse: 5′-TTGAGGTAGGTCTGGTGAAGGTCC-3′; IL-6 forward: 5′-CCCCTGACCCAACCACAAAT-3′ and reverse: 5′-ATTTGCCGAAGAGCCCTCAG-3′, IL-8 forward: 5′-GAGTGGACCACACTGCGCCA-3′ and reverse: 5′-TCCACAACCCTCTGCACCCAGT-3′.

### 4.5. Western Blot Analysis

Western blotting was performed, as described previously [49]. Antibodies used in this study are as follows: IL-6 (Cell Signaling Technology, Danvers, MA, USA, D3K2N), IL-6 (Immunoway, YT5348), IL-8 (Cell Signaling Technology, E5F5Q), H3 (Abcam, Cambridge, UK, ab1971), H3K9me2 (Cell Signaling Technology, D85B4), H3K9me3 (Millipore, Burlington, MA, USA, #07-523), H3K27me3 (Active Motif, Carlsbad, CA, USA, 61017), H3K27me3 (Millipore, #07-449), p-STING (Cell Signaling Technology, 19781), STING (Abways, Shanghai, China, CY7204), c-GAS (Cell Signaling Technology, 79978), c-GAS (Abcam, ab242363), β-actin (Sigma-Aldrich, St. Louis, MO, USA, A5228), CyclinA2 (Abcam, ab181591), CyclinA2 (Cell Signaling Technology, E6D1J), and Ki67 (GeneTex, Irvine, CA, USA, GTX16667). Secondary goat anti-mouse and goat anti-rabbit antibodies were obtained from ZSGB-BIO (Beijing, China).

### 4.6. Immunofluorescence

Immunofluorescence was performed, as described previously [49].

### 4.7. Wound Healing Assays

Wound healing was performed, as described previously [49].

### 4.8. Flow Cytometric Analysis

Cell cycle assay was performed, as described previously [49].

### 4.9. SA-β-Gal Staining

SA-β-gal staining was performed, as described previously [49].

### 4.10. Cell Viability Assays

To assess the antiproliferative effects of the EZH2 and G9a inhibitors, tumor cell lines (typically 1 × 10^3^ cells/100 μL/well) were plated in triplicate in 96-well plates and incubated with fresh medium for 24 h. Then, the cells were incubated with a single inhibitor (UNC1999 or UNC0642) or a combination of the two inhibitors (UNC1999 + UNC0642) or with DMSO for up to seven days, refreshing the medium and drugs every three days. Cell viability was assessed on days one, three, five, and seven using the Cell Counting Kit-8 (CCK8) (APE, k1018) or MTT Assay Kit (Abcam, ab211091) and a microplate reader (Tecan Infinite M NANO+, Austria), according to the manufacturer’s instructions.

### 4.11. Statistical Analysis

The results were compiled from at last three independent replicate experiments and are presented as the mean ± SD. The unpaired Student’s *t*-test (two-tailed) was used to calculate the significance of differences between groups. Data were considered significant at *p* < 0.05 (* *p* < 0.05, ** *p* < 0.01, *** *p* < 0.001). Statistical analysis was performed using GraphPad Prism software (GraphPad Software, La Jolla, CA, USA).

## 5. Conclusions

In conclusion, we demonstrated that the combination of an EZH2 inhibitor and a G9a inhibitor induced senescence in highly metastatic tumor cells by targeting H3K9me2 and H3K27me3 and without triggering SASP, providing a potentially promising epigenetics-based therapeutic strategy for the treatment of highly metastatic tumors. Moreover, a combination of EZH2 inhibitor, G9a inhibitor, and other chemotherapy drugs may be an effective therapeutic strategy for the treatment of tumors in the future.

## Figures and Tables

**Figure 1 ijms-23-03911-f001:**
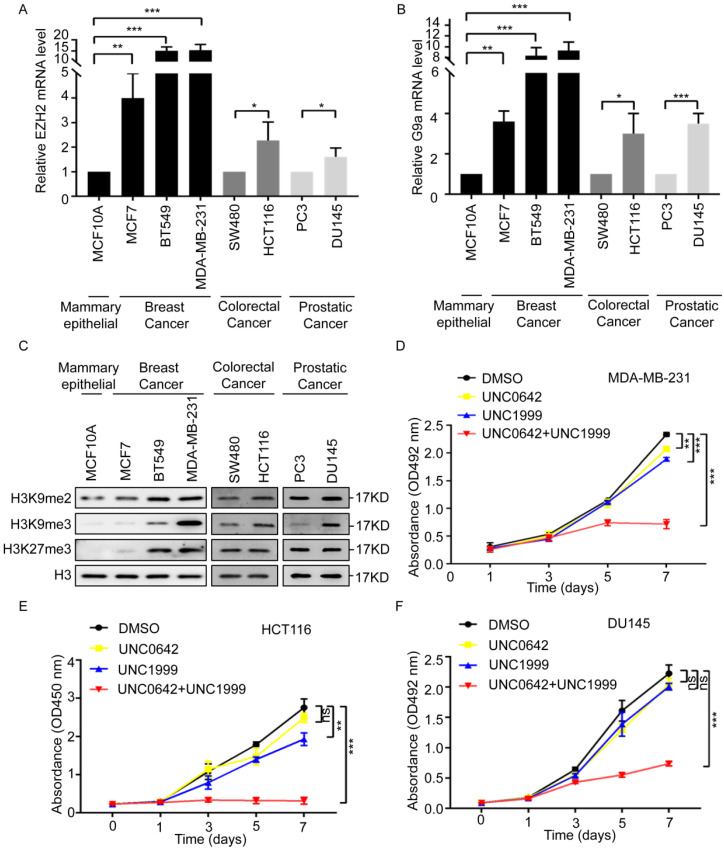
A combination of UNC0642 and UNC1999 inhibits the proliferation and migration of tumor cells. (**A**) qRT-PCR analysis of the expression of EZH2 in noncancerous mammary epithelial cell (MCF10A) and breast cancer cells (MCF7, BT549, and MDA-MB-231), colon cancer cells (SW480 and HCT116) and prostate cancer cells (PC3 and DU145). (**B**) qRT-PCR analysis of the expression of G9a in noncancerous mammary epithelial cell (MCF10A) and breast cancer cells (MCF7, BT549, and MDA-MB-231), colon cancer cells (SW480 and HCT116), prostate cancer cells (PC3 and DU145). (**C**) Western blot analysis of expression of H3K9me2, H3K9me3, and H3K27me3 in noncancerous mammary epithelial cell (MCF10A) and breast cancer cells (MCF7, BT549, and MDA-MB-231), colon cancer cells (SW480 and HCT116), and prostate cancer cells (PC3 and DU145). (**D**–**F**) MTT or CCK8 assay was performed to detect the proliferation of MDA-MB-231 cells (**D**), HCT116 cells (**E**), and DU145 cells (**F**) treated with indicated treatments for seven days. Each experiment was repeated at least three times. Error bars, mean ± SD, ns: no significant, * *p* < 0.05, ** *p* < 0.01, *** *p* < 0.001.

**Figure 2 ijms-23-03911-f002:**
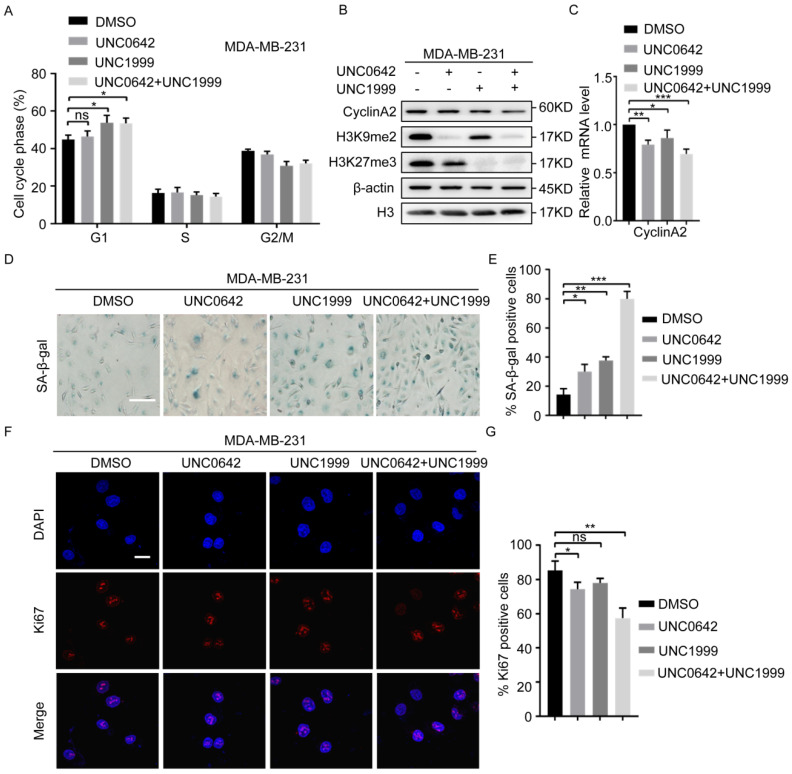
A combination of UNC0642 and UNC1999 induces MDA-MB-231 cellular senescence. (**A**) Cell cycle analysis in MDA-MB-231 cell treated with indicated treatments. (**B**) Western blot analysis of expression of CyclinA2, H3K9me2, and H3K27me3 in MDA-MB-231 cells treated with indicated treatments. (**C**) qRT-PCR analysis of the expression of CyclinA2 in MDA-MB-231 cells treated with indicated treatments. (**D**,**E**) SA-β-Gal staining in MDA-MB-231 cells treated with indicated treatments. Left panel represents images of SA-β-Gal (**D**), and right panel represents the statistical analysis of the percentage of SA-β-gal positive cells, scale bar: 100 nm (**E**). (**F**,**G**) Immunofluorescence analysis of Ki67 in MDA-MB-231 cells with the indicated treatments. Representative images of Ki67 positive cells were showed (**F**) and the percentage of the Ki67 positive cells was calculated, scale bar: 20 μm (**G**). Each experiment was repeated at least three times. Error bars, mean ± SD, ns: no significant, * *p* < 0.05, ** *p* < 0.01, *** *p* < 0.001.

**Figure 3 ijms-23-03911-f003:**
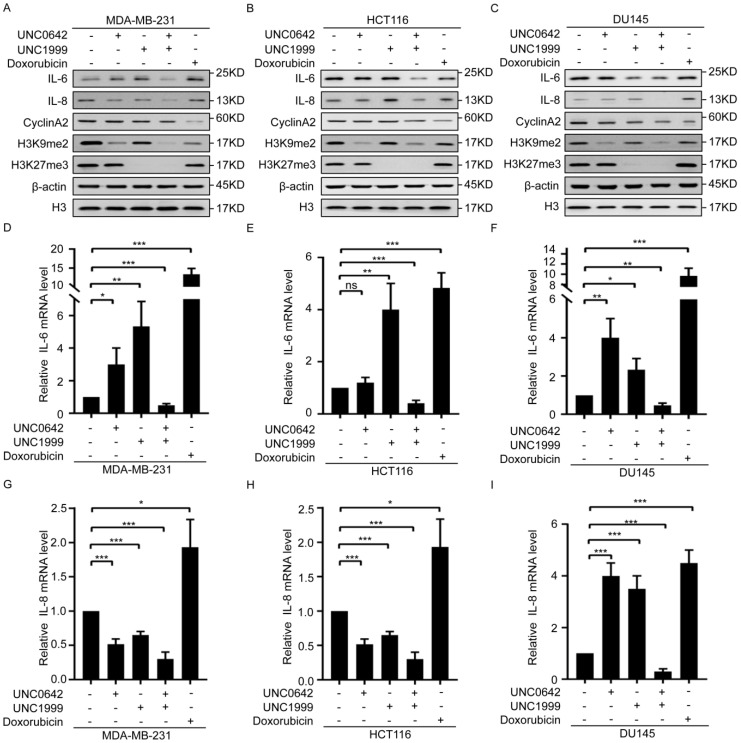
A combination of UNC0642 and UNC1999 inhibits the secretion of SASP. (**A**–**C**) Western blot analysis of expressions of IL-6, IL-8, CyclinA2, H3K9me2, and H3K27me3 in MDA-MB-231 cells (**A**), HCT116 cells (**B**), and DU145 cells (**C**) treated with indicated treatments. (**D**–**F**) qRT-PCR analysis of the expression of IL-6 in MDA-MB-231 cells (**D**), HCT116 cells (**E**), and DU145 cells (**F**) treated with indicated treatments. (**G**–**I**) qRT-PCR analysis of the expression of IL-8 in MDA-MB-231 cells (**G**), HCT116 cells (**H**), and DU145 cells (**I**) treated with indicated treatments. Each experiment was repeated at least three times. Error bars, mean ± SD, ns: no significant, * *p* < 0.05, ** *p* < 0.01, *** *p* < 0.001.

**Figure 4 ijms-23-03911-f004:**
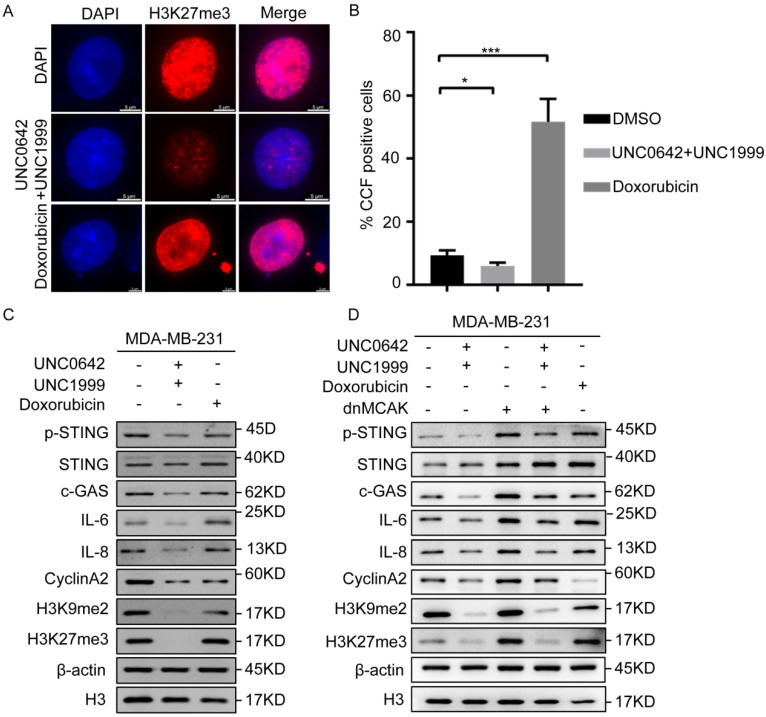
A combination of UNC0642 and UNC1999 inhibits cGAS-STING signaling by decreasing the formation of CCF. (**A**,**B**) Immunofluorescence analysis of H3K27me3, a marker of CCF in MDA-MB-231 cells with the indicated treatments. Representative images of CCF positive cells were shown (**A**), and the percentage of the CCF positive cells were calculated, scale bar: 5 μm (**B**). (**C**,**D**) Western blot analysis of expression of p-STING, STING, c-GAS, IL-6, IL-8, CyclinA2, H3K9me2, and H3K27me3 in MDA-MB-231 cells treated with indicated treatments. Each experiment was repeated at least three times. Error bars, mean ± SD, * *p* < 0.05, *** *p* < 0.001.

**Figure 5 ijms-23-03911-f005:**
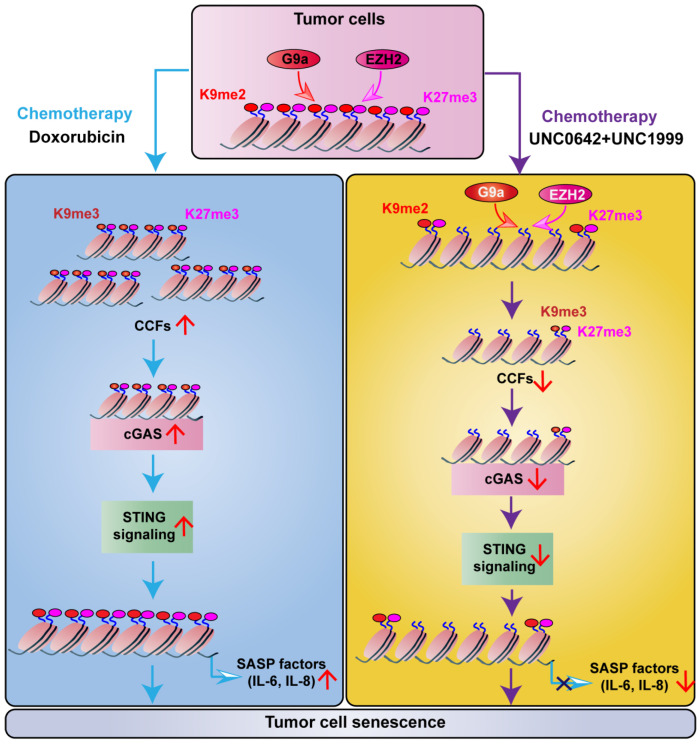
Work model. In tumor cells, doxorubicin induces cell senescence, as indicated by the increased formation of CCF. Thereby, CCF targets and activates the cGAS-STING-SASP, causing the secretion of IL-6 and IL-8. Meanwhile, a combination of UNC0642 and NC1999 inhibits the functions of G9a and EZH2, causing the decreasing levels of H3K9me3 and H3K27me3. Moreover, dual inhibition of H3K9me2 and H3K27me3 promotes tumor cell senescence without triggering SASP by decreasing the formation of CCF, which is dependent on the inhibition of cGAS-STING signaling.

## Data Availability

The data presented in this study are available in the Appendix A.

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
