# Peer review of "Dual Inhibition of H3K9me2 and H3K27me3 Promotes Tumor Cell Senescence without Triggering the Secretion of SASP"

_ijms, 2022, doi:10.3390/ijms23073911_

Round 1
Reviewer 1 Report
The paper of Zhang et al. is well written, well designed and the results are convincing, therefore it deserves to be published in IJMS.
I only have a few remarks that should be noted before publication, especially regarding the presentation of the data, here are my comments:
INTRODUCTION
Move the concept expressed in lines 70-74 to the conclusion and leave the ending of the introduction with the aim of the study
RESULTS
Figures 1A-1B-1C cell line details are not aligned with the graph above
For figure 2F (immunofluorescence) the merge of DAPI and Ki67 should be shown; furthermore, better representative images should be provided based on the graph shown
Figures 3A-3B-3C there would seem to be a double band (very close each other) for the IL-6 WB; given the proximity of the bands how the authors distinguished which was IL-6? I recommend replacing the images with more representative ones
Figure 3C, the H3 WB should be better cut (or replaced) since there would seem to be a smudge at the beginning (maybe it was the marker well?)
CONCLUSION
I also suggest adding a concluding paragraph (5. Conclusion), in which to move the lines 298-301 of the conclusion, also adding possible future perspectives from a translational perspective.
Author Response
INTRODUCTION
Move the concept expressed in lines 70-74 to the conclusion and leave the ending of the introduction with the aim of the study.
Response: We appreciated the reviewer’s suggestion. As shown in the revised manuscript, we have modified the text according to the reviewer's suggestion.
RESULTS
- Figures 1A-1B-1C cell line details are not aligned with the graph above.
Response: We have rearranged the figures to make the cell line details be aligned with the graph.
- For figure 2F (immunofluorescence) the merge of DAPI and Ki67 should be shown; furthermore, better representative images should be provided based on the graph shown.
Response: As suggested we have provided better representative images and merged of DAPI and Ki67 in figure 2F and other figures (immunofluorescence).
- Figures 3A- there would seem to be a double band (very close each other) for the IL-6 WB; given the proximity of the bands how the authors distinguished which was IL-6? I recommend replacing the images with more representative ones.
Response: A double band appeared due to the specificity of IL-6 antibody. Though the proximity of the bands, the expression trend of IL-6 is consistent and it doesn’t affect the conclusion. We have replaced the images with more representative ones as reviewer’s suggestion.
- Figure 3C, the H3 WB should be better cut (or replaced) since there would seem to be a smudge at the beginning (maybe it was the marker well?)
Response: The smudge at the beginning is the marker well. We have re-cut the H3 WB.
CONCLUSION
I also suggest adding a concluding paragraph (5. Conclusion), in which to move the lines 298-301 of the conclusion, also adding possible future perspectives from a translational perspective.
Response: As suggested, we have added a concluding paragraph (5. Conclusion) in the revised manuscript.
Reviewer 2 Report
Overall summary
The authors conducted a study to examine the ability of 2 small molecules inhibitors UNC0642 and UNC1999 to inhibit H3K9 and H3K27 methylation in breast, colon and prostate cancer cells. They observed that the inhibition of H3K9me2 and H3K27me3 had a stronger pro-senescence effect than either inhibitor alone without triggering senescence-associated secretory phenotype (SASP) in highly metastatic tumor cells (MDA-MB-231) by inhibiting the cGAS-STING-SASP pathway.
General comments
The article is interesting, and it is quite well written. Furthermore, this manuscript could have great clinical, social and academic relevance. However, there are deficiencies which should be addressed before publication could be considered.
- Some parts of the introduction are very similar to their original (see document attached). Authors must re-write them. In general, there should not be consecutive 6 words copied. You can use duplication check websites to inspect your manuscript before you send it back.
- Why MDA-MB-231 cells were used to determinate the optimum concentration of inhibitors assayed?
- References: The references list must be reviewed. For example, abbreviated journal names must be used (see https://www.mdpi.com/journal/ijms/instructions).
Other comments
- The abbreviations used in Figure S1 (BRCA, COAD…) must be clarified in the legend.
- Line 162. The promotion of senescence by inhibiting H3K9me2 and H3K27m3 is shown only in Figure 2 (not Figure 1 and 2)
- Line 215. The inhibition of the formation of CCFs by inhibiting H3K9me2 and H3K27m3 is shown in Figure 4B (not Figure 4C).
- Line 254. The induction of senescence by inhibiting H3K9me2 and H3K27m3 is shown in Figure 2 (not Figure 1).
- Line 256: Change MCF by MCF7
- Uniformity in the terminology is recommended: HEK293T or HEK-293T

Author Response
General comments
- Some parts of the introduction are very similar to their original (see document attached). Authors must re-write them. In general, there should not be consecutive 6 words copied. You can use duplication check websites to inspect your manuscript before you send it back.
Response: We thank the reviewer for pointing this out and we have re-written parts of the introduction in the revised manuscript.
- Why MDA-MB-231 cells were used to determinate the optimum concentration of inhibitors assayed?
Response: We thank the reviewer for raising this point. In fact, we used two breast cancer cell lines (MDA-MB-231 and MCF7) to determine the optimum concentration of the inhibitors. We have added the data of MCF7 in the Figure S2. In addition, the optimal concentrations of the inhibitors are also suitable for the colon cancer cells and the prostate cancer cells used.
- References: The references list must be reviewed. For example, abbreviated journal names must be used (see https://www.mdpi.com/journal/ijms/instructions).
Response: We are sorry for the errors. We have corrected them by modifying the text.
Other comments
- The abbreviations used in Figure S1 (BRCA, COAD…) must be clarified in the legend.
Response: We have clarified the abbreviations in the legend.
- Line 162. The promotion of senescence by inhibiting H3K9me2 and H3K27m3 is shown only in Figure 2 (not Figure 1 and 2)
Response: We thank the reviewer for pointing this out. “Our initial experiments showed that dual inhibition of H3K9me2 and H3K27me3 promotes tumor cell senescence (Figure 1 and 2)” should be “Our initial experiments showed that dual inhibition of H3K9me2 and H3K27me3 promotes tumor cell senescence (Figure 2)”. We have corrected them in the revised manuscript.
- Line 215. The inhibition of the formation of CCFs by inhibiting H3K9me2 and H3K27m3 is shown in Figure 4B (not Figure 4C).
Response: We have corrected them in the revised manuscript. “Because our initial results showed that dual inhibition of H3K9me2 and H3K27me3 inhibited the formation of CCF (Figure 4C)” should be “Because our initial results showed that dual inhibition of H3K9me2 and H3K27me3 inhibited the formation of CCF (Figure 4B)”
- Line 254. The induction of senescence by inhibiting H3K9me2 and H3K27m3 is shown in Figure 2 (not Figure 1).
Response: We have corrected them in the revised manuscript. “the effect was stronger than that of either inhibitor alone (Figure 1 and Figure S4)” should be “the effect was stronger than that of either inhibitor alone (Figure 2 and Figure S4)”.
- Line 256: Change MCF by MCF7
Response: We have corrected it by modifying the text. “MCF” should be “MCF7”.
- Uniformity in the terminology is recommended: HEK293T or HEK-293T
Response: We have uniformly modified to “HEK293T” in the revised manuscript.
Round 2
Reviewer 1 Report
The authors replied well, the manuscript is now significantly improved, so for me it should be accepted.